

# Topological lattice models with constant Berry curvature

**Daniel Varjas⋆, Ahmed Abouelkomsan, Kang Yang and Emil J. Bergholtz**

Department of Physics, Stockholm University, AlbaNova University Center,
106 91 Stockholm, Sweden

⋆ dvarjas@gmail.com

## Abstract

Band geometry plays a substantial role in topological lattice models. The Berry curvature, which resembles the effect of magnetic field in reciprocal space, usually fluctuates throughout the Brillouin zone. Motivated by the analogy with Landau levels, constant Berry curvature has been suggested as an ideal condition for realizing fractional Chern insulators. Here we show that while the Berry curvature cannot be made constant in a topological two-band model, lattice models with three or more degrees of freedom per unit cell can support exactly constant Berry curvature. However, contrary to the intuitive expectation, we find that making the Berry curvature constant does not always improve the properties of fractional Chern insulator states. In fact, we show that an "ideal flatband" cannot have constant Berry curvature, equivalently, we show that the density algebra of Landau levels cannot be realised in any tight-binding lattice system.


# 1 Introduction

Landau levels (LL) arise in a two-dimensional (2D) electron gas under strong magnetic field. The kinetic energy is frozen inside each LL and the topological character of single-electron states leads to the integer quantum Hall effect (IQHE) [1] when a LL is completely filled. At fractional fillings, the system is dominated by the electron-electron interaction and the fractional quantum Hall effect (FQHE) [2–5] can take place. These phases have attracted much attention in the past decades, due to the potential applications of their anyonic excitations as building blocks of a topological quantum computer [6–8]. While quantum Hall physics originates from the LL structure in 2D continuum, many of its characteristic aspects are also reproduced in lattice models with discrete translational symmetry. The key ingredient of IQHE lies in the band geometry [9], characterized by the Berry curvature. The Berry curvature acts analogously to an external magnetic field, but in momentum space, and has several applications in transport calculations [10,11]. Chern insulators [12], which host bands with nontrivial Berry curvature whose integral is quantized to the Chern number, display the quantized conductance and topological edge states associated with the IQHE.

Due to the similarity between Chern bands and LLs, an analogue of the FQHE state appears when the bandwidth is small compared to the interaction scale and a Chern band is partially filled: the fractional Chern insulator (FCI) [13–15]. While FCI states have been experimentally realized in the presence of weak external magnetic field [16], there has been no experimental realization yet in the absence of any external magnetic field. Much effort has been invested in looking for conditions stabilizing these FCIs [17–19]. Intuitively, one would expect that the more the band structure is similar to a LL, the more robust the FCI states are. A LL has completely flat dispersion and the projected density operator satisfies the Girvin-Macdonald-Platzman (GMP) algebra [20]. The possibility of realizing these properties has attracted further interest since the discovery of topological flatbands in Moiré systems [21–29].

All this raises the natural questions: can these LL properties be exactly reproduced in a lattice system? If so, how do they stabilize the FCI states? Ref. [30] points out the negative result that an exactly flat Chern band with no dispersion cannot be realized for finite-range hoppings. Besides the energy dispersion, band structures are characterized geometrically by the Berry curvature and the Fubini-Study metric, which constitute the real and imaginary parts of the quantum geometric tensor defined in sec. 2. The role of the Berry curvature has been well understood in quantum Hall physics, and it has been shown through numerical studies that there is a correlation between the stability of FCI states and Berry curvature fluctuations in a number of lattice models [31]. This motivated a search for bands with as flat as possible energy dispersion, and Berry curvature with as small as possible variations [13,15,18,32,33]. The Fubini-Study metric has been recently identified to play a role in the collective mode of FQHE [34,35], and Ref. [18] showed that the GMP algebra is recovered in a Chern band with constant Berry curvature and constant Fubini-Study metric saturating a certain inequality.

In this manuscript we ask the basic question: is it possible to construct bands with a Berry curvature that is exactly constant? We answer this question by providing a construction to obtain constant curvature bands in models with three or more bands (sec. 3 and Fig. 1), and proving that this is impossible in 2-band models (sec. 4). Next, we investigate the consequences of constant curvature on the physics of FCI states in such bands (sec. 5). We find that minimizing curvature variations does not generally make the FCI state more "ideal". The key property that governs the degeneracy pattern of the FCI droplet is the relation between the Berry curvature and the Fubiny-Study metric. We show that this relation cannot be satisfied while keeping the curvature constant (sec. 6). This is equivalent to the fact that the exact GMP density algebra cannot be reproduced in a lattice system with finite number of degrees of freedom per unit cell.

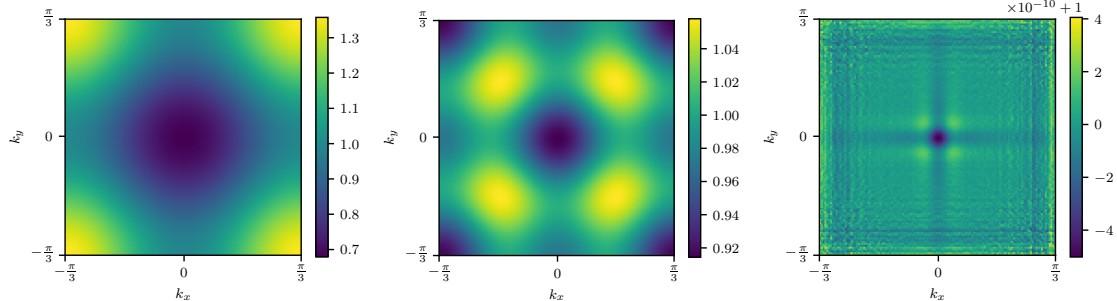

Figure 1: Berry curvature of the 3-band Kapit-Mueller model (left), after one iteration of the flattening algorithm (middle), and after 12 iterations (right). Note the scales of the colorbars. The curvature is scaled such that average curvature of 1 corresponds to a band with Chern number 1. Because of the magnetic translation symmetry by one lattice constant, the Berry curvature pattern repeats three times, and we only show one third of the magnetic Brillouin zone.

## 2 Band geometry in tight-binding models

The non-interacting band structure of a translation-invariant tight-binding system is characterized by the $n \times n$ Bloch Hamiltonian $H(\mathbf{k})$, where $n$ is the number of orbitals inside each unit cell, and its normalized eigenstates $u_{\mathbf{k}}^{(m)}$ with $m$ as the band index. The quasimomentum $\mathbf{k}$ takes values in the Brillouin zone (BZ) corresponding to the magnetic unit cell that has integer magnetic flux penetrating it. In the following we study properties of a single occupied band, and drop the band index $m$.

When the unit cell has more than one site at different spatial coordinates, it is conventional to use the *periodic gauge* of Bloch states [10]. This basis simplifies calculations of electromagnetic properties, correctly taking the real-space structure into account. The boundary condition for the Bloch Hamiltonian in this basis is

$$H(\mathbf{k} + \mathbf{G}) = W_{\mathbf{G}} H(\mathbf{k}) W_{-\mathbf{G}}, \tag{1}$$

where $W_{\mathbf{G}} = \exp(i\mathbf{G} \cdot \mathbf{r})$ with $\mathbf{G}$ a reciprocal lattice vector and $\mathbf{r}$ the position operator. This is a diagonal operator in the basis of the localized tight-binding orbitals, $(W_{\mathbf{G}})_{ij} = \delta_{ij} \exp(i\mathbf{G} \cdot \mathbf{r}_i)$ where $i, j$ index the $n$ orbitals of the unit cell and $\mathbf{r}_i$ is the real space position of orbital $i$. The wavefunction obeys the boundary condition

$$u_{\mathbf{k}+\mathbf{G}} = W_{\mathbf{G}} u_{\mathbf{k}}. \tag{2}$$

These boundary conditions can also be interpreted as the prescription to extend $H(\mathbf{k})$ and $u_{\mathbf{k}}$ from the first BZ to $\mathbb{R}^2$.

The geometrical properties of the band are characterized by the *quantum geometric tensor*

$$\eta_{\mu\nu}(\mathbf{k}) = g_{\mu\nu}(\mathbf{k}) + \frac{i}{2}\epsilon_{\mu\nu}\mathcal{F}(\mathbf{k}) = \left(\partial_\mu u_{\mathbf{k}}^\dagger\right)\left(\mathbb{1} - u_{\mathbf{k}} u_{\mathbf{k}}^\dagger\right)(\partial_\nu u_{\mathbf{k}}), \tag{3}$$

where $\mu, \nu$ index spatial directions $x, y$, $\partial_\mu = \partial/\partial k_\mu$, $\epsilon_{\mu\nu}$ is the antisymmetric tensor, and we introduced the decomposition into the real symmetric *Fubini-Study metric g* and the scalar *Berry curvature* $\mathcal{F}$. The *Chern number* is a quantized topological invariant proportional to the Hall conductivity, given by the integral of the Berry curvature over the BZ, $C = \frac{1}{2\pi} \int_{\mathrm{BZ}} \mathcal{F}$. For topological bands with nonzero Chern number, we need to interpret $u_{\mathbf{k}}$ as a mapping to the complex projective space $\mathbb{C}P^{n-1}$, as it cannot be a global section of $\mathbb{C}^n$, (2) is only

satisfied up to an overall complex phase for the wavefunction in $\mathbb{C}^n$. The geometrical properties of the band are insensitive to changing the wavefunction by a $\mathbf{k}$-dependent overall complex phase, and (3) is well defined both for a gauge-fixed normalized wavefuntion in $\mathbb{C}^n$ or the wavefunction in $\mathbb{C}P^{n-1}$. It should be noted, however, that these properties, with the exception of the Chern number, depend on the embedding in real-space, i.e. on the spatial structure of the unit cell [36].

## 3 General method to make the curvature constant

In this section we provide an algorithm to construct a Bloch Hamiltonian with constant Berry curvature, through a deformation of any Hamiltonian with nonzero curvature. We start with a Hamiltonian $H(\mathbf{k})$ and replace it with $H'(\mathbf{k}) = H(\mathbf{f}(\mathbf{k}))$ where $\mathbf{f}$ is a smooth, periodic function mapping the BZ to itself. If $H$ had a Berry curvature $\mathcal{F}$ it transforms into

$$\mathcal{F}'(\mathbf{k}) = \mathcal{F}(\mathbf{f}(\mathbf{k})) \det\left(\frac{d\mathbf{f}}{d\mathbf{k}}\right), \tag{4}$$

because it transforms as a volume form. According to Moser's theorem [37] a deformation with $\mathcal{F}'(\mathbf{k}) = $ const. exists for any smooth $\mathcal{F}$ that does not have any zeros.

To get an approximate solution, let us assume that the curvature is already almost constant, so with proper normalization it can be written as $\mathcal{F}(\mathbf{k}) = 1 + \epsilon(\mathbf{k})$ with $|\epsilon(\mathbf{k})| \ll 1$. The transformation we are looking for is $\mathbf{f}(\mathbf{k}) = \mathbf{k} + \mathbf{h}(\mathbf{k})$ with small $\mathbf{h}$. We can expand the determinant as

$$\det\left(\frac{d\mathbf{f}}{d\mathbf{k}}\right) \approx 1 + \mathrm{tr}\left(\frac{d\mathbf{h}}{d\mathbf{k}}\right). \tag{5}$$

Choosing

$$\mathrm{tr}\left(\frac{d\mathbf{h}}{d\mathbf{k}}\right) \equiv \nabla_{\mathbf{k}} \cdot \mathbf{h}(\mathbf{k}) = -\epsilon(\mathbf{k}) \tag{6}$$

the curvature $\mathcal{F}'$ is 1 up to second order in $\epsilon$ and its derivatives. This is accomplished by using the Fourier series (using $\mathbf{x}$ as the reciprocal coordinate of $\mathbf{k}$) and setting

$$\mathbf{h}(\mathbf{x}) = i \frac{\mathbf{x}}{|\mathbf{x}|^2} \epsilon(\mathbf{x}). \tag{7}$$

In our numerical implementation we sample $\mathbf{k}$ and $\mathbf{x}$ on a discrete $N \times N$ grid, and use the inverse of the discrete divergence operator, replacing $|\mathbf{x}|^2$ in the denominator of (7) with $N/(2\pi)\mathbf{x} \cdot \sin(2\pi\mathbf{x}/N)$.

This transformation of $\mathcal{F} \to \mathcal{F}'$ can be iterated until the desired flatness is reached. Finding the exact conditions for the convergence of this algorithm is outside of the scope of this manuscript, but we find that the algorithm converges quickly for the smooth functions that we encounter in our test cases.

Smooth Bloch Hamiltonians $H(\mathbf{k})$ correspond to tight-binding Hamiltonians in real space with hopping matrix elements decaying exponentially. The above deformation maintains the smoothness of the Hamiltonian, the resulting $H'$ remains exponentially localized in real space. Moreover, the new energy spectrum is $E'(\mathbf{k}) = E(\mathbf{f}(\mathbf{k}))$, hence the flatness of bands is unaffected.

We numerically demonstrate that the above flattening procedure results in lattice models with almost constant Berry curvature to arbitrary precision. We use three and four-band models with both different and the same positions of the orbitals within the magnetic unit cell. Similar constructions work for any $N \geq 3$ number of bands. We start from the Kapit-Mueller (KM) Hamiltonian [32] with $\phi = 1/3$ flux per plaquette and three sites in the magnetic unit

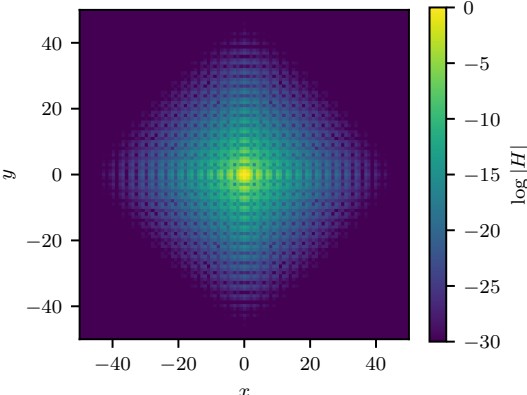

Figure 2: Magnitude of hopping matrix elements in the modified KM Hamiltonian with constant Berry curvature, as function of relative site positions.

cell. The lowest band in this model is an exact flat band and the Berry curvature is positive everywhere in the BZ. In the numerical calculations we truncate the KM model to tenth neighbor hoppings, further neighbor hoppings have relative amplitude under $10^{-8}$ and do not significantly change the band structure.

Applying the flattening iteration described above 12 times, the Berry curvature becomes constant within $10^{-9}$ relative variation, see Fig. 1. After this point, numerical noise starts to dominate the variations of the curvature, and further iterations do not improve the result. After the optimization procedure to minimize fluctuations of the Berry curvature, the resulting Hamiltonian still is exponentially localized in real space, see Fig. 2.

We also apply the optimization algorithm to the four-band ($\phi = 1/4$ flux per plaquette) Hofstadter model with Chern number $C = 1$ in the lowest band. The resulting Berry curvature has relative variations of order $10^{-6}$, as shown in Fig. 3. Furthermore, we demonstrate the algorithm on the 3-band model of Ref. [38], which has Chern number $C = 3$, the results are shown in Appendix A.

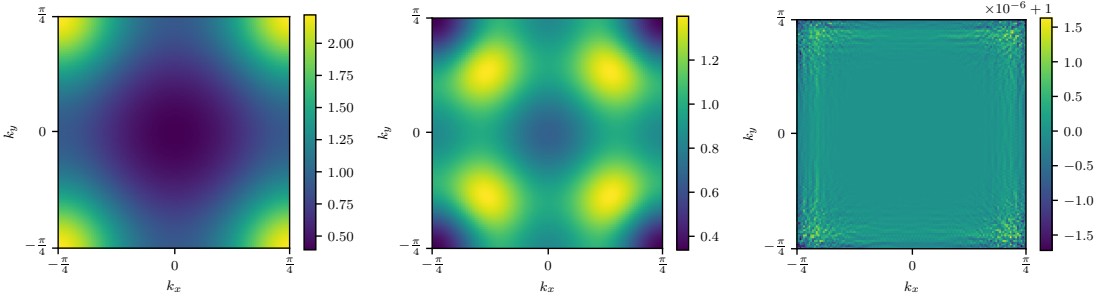

Figure 3: Berry curvature of the 4-band ($\phi = 1/4$) Hofstadter model (left), after one iteration of the flattening algorithm (middle), and after 19 iterations (right). Note the scales of the colorbars. The curvature is scaled such that average curvature of 1 corresponds to a band with Chern number 1. Because of the magnetic translation symmetry by one lattice constant, the Berry curvature pattern repeats four times, and we only show one fourth of the BZ.

# 4 No-go theorem in two-band models

Before moving on to study FCI physics in constant curvature bands, we prove a no-go theorem: in two-band models the fluctuations of the Berry curvature have a finite lower bound, hence constant curvature is impossible. This may be a reason why such band structures eluded discovery so far. This result has also been proved recently in the case of a single site per unit cell [39]. Here we give a more detailed proof and generalize the statement to systems where the unit cell has spatial structure so that the Bloch Hamiltonian is not necessarily periodic in reciprocal lattice vectors.

We first look at the case when all the orbital positions inside a unit cell coincide with the lattice sites. In this situation, we can view the Chern band as a map from the torus $T^2$ to the Bloch sphere $S^2 \simeq \mathbb{C}P^1$, which is denoted as $p$. The Berry curvature has the geometric meaning of the solid angle on the Bloch sphere, $|\mathcal{F}| dk_x dk_y = d\Omega$. If the Berry curvature is non-vanishing everywhere, then the map $p$ is a local diffeomorphism according to the inverse function theorem. From the local diffeomorphism, we can deduce that the image $p(T^2)$ is open in $S^2$. On the other hand, as $T^2$ is compact, $p(T^2)$ is also compact and thus closed in $S^2$. So $p$ is a surjection from $T^2$ to $S^2$. For each point $x$ on $S^2$, we denote its preimage as $p^{-1}(x)$. Since $x$ is closed, the preimage $p^{-1}(x)$ is closed and therefore compact in $T^2$. On the other hand, for each point $y_i \in p^{-1}(x)$, the local diffeomorphism tells us that there is an open neighbourhood $U_i$ of $y_i$ which does not contain other preimages of $x$. These $U_i$ form an open cover of $p^{-1}(x)$ and can only be a finite set due to the compactness. As a result, we can choose an open neighborhood $\bigcap_i p(U_i)$ of $x$ which is evenly covered by $p$. In this case, we have a covering map from $T^2$ to $S^2$. A covering map induces an injective map for the homotopy group $\pi_1$ [40]. However, the homotopy group $\pi_1$ of the torus is $\mathbb{Z} \times \mathbb{Z}$ while $\pi_1$ of the sphere is trivial, leading to a contradiction. This shows that $\mathcal{F}$ must vanish somewhere in the Brillouin zone.

If the site positions are all rational multiples of the unit vectors, there are reciprocal lattice vectors $\tilde{\mathbf{G}}_i$ such that $W_{\tilde{\mathbf{G}}_i} = \mathbb{1}$. These define an extended Brillouin zone where the wavefunction is periodic. The Berry curvature is the same in every copy of the first BZ, because a constant unitary transformation does not change the curvature. So the Chern number is also nonzero in the extended BZ, and the no-go theorem for 2-band models with BZ periodic wavefunctions applies, meaning that the curvature has to vanish somewhere.

The Berry curvature $\mathcal{F}$ is a continuous function of the components of the position operator $\mathbf{r}$. As $\int \mathcal{F} = 2\pi C$, $\max \mathcal{F} \geq 2\pi C/A$ where $A$ is the area of the BZ. Since for rational $\mathbf{r}$ we know $\mathcal{F}$ must vanish somewhere, we have $\max \mathcal{F} - \min \mathcal{F} \geq 2\pi C/A$ (we assume $\max \mathcal{F}$ positive) at rational $\mathbf{r}$. As $\mathcal{F}$ is a continuous function of the site positions (keeping the onsite and hopping terms in the tight-binding model constant), it is not hard to show that $\max \mathcal{F}$ and $\min \mathcal{F}$ are also continuous based on the compactness of BZ. So $\max \mathcal{F} - \min \mathcal{F} \geq 2\pi C/A$ is also satisfied for irrational positions. Thus, the Berry curvature cannot be uniform even if we deform the position of the sublattice sites.

# 5 Fractional Chern insulators with constant curvature

In this section, we test the expectation that fractional Chern insulator states are more stable in flatbands with smaller Berry curvature variations. While this might hold in some cases, we argue here that it is not generally true. We demonstrate this by studying bosonic FCI states in the modified KM model with constant Berry curvature defined in section 3.

The original KM model has the remarkable property that its lowest-band eigenstates are lattice versions of the lowest LL wave functions [32]. On the torus, this implies the existence of two exact zero modes in the many-body spectrum at half filling for on-site interactions, since

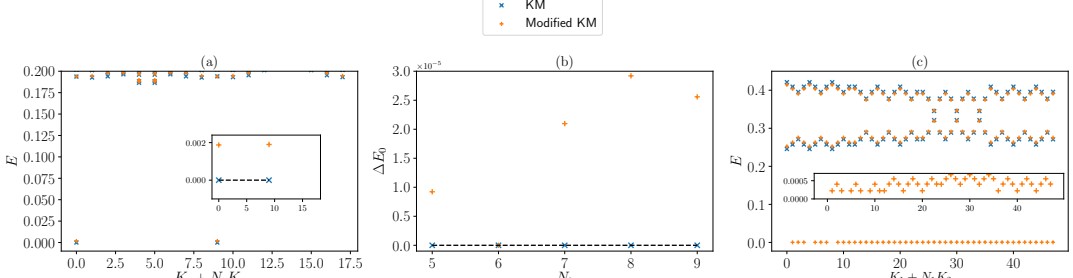

Figure 4: (a) The many-body spectrum at half filling for a 3-band ($\phi = 1/3$) KM model and its modified version with the constant Berry curvature for a lattice of $N_1 \times N_2/\phi = 9 \times 8$ sites. The inset shows the two-fold degenerate ground states. The dashed horizontal line is the zero energy line. (b) The two-fold ground state degeneracy splitting $\Delta E_0$ at half filling for different systems of $N_b$ bosons in a $N_b \times 2/\phi$ lattice. The dashed horizontal line is the zero ground state splitting line. When the lattice is perfectly square ($N_b = 6$), the modified model has the same ground state energies as the original model hence the same ground state splitting. (c) The two-body spectrum for a system of $N_1 \times N_2 = 12 \times 4$ magnetic unit cells. The inset shows the lowest non-zero two-body energy per total momentum sector for the modified KM model.

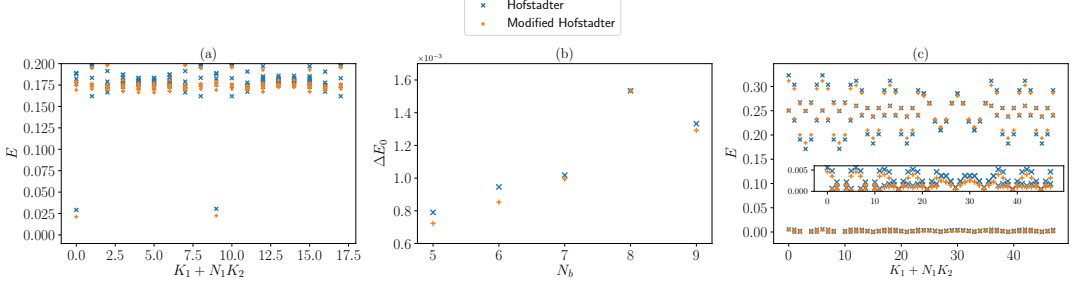

Figure 5: (a) The many-body spectrum at half filling for a 4-band ($\phi = 1/4$) Hofstadter model and its modified version with the constant Berry curvature for a lattice of $N_1 \times N_2/\phi = 9 \times 8$ sites. (b) The two-fold ground state degeneracy splitting $\Delta E_0$ at half filling for different systems of $N_b$ bosons in a $N_b \times 2/\phi$ lattice. When the lattice is perfectly square ($N_b = 8$), the modified model has the same ground state energies as the original model hence the same ground state splitting. (c) The two-body spectrum for a system of $N_1 \times N_2 = 12 \times 4$ magnetic unit cells. The inset shows the lowest non-zero two-body energies per total momentum sector for both models.

bosonic Laughlin states are exact zero modes of a parent Hamiltonian described by contact interactions $V(\mathbf{r}_1, \mathbf{r}_2) = V_0 \delta^2(\mathbf{r}_1 - \mathbf{r}_2)$ in the continuum. Exact zero modes have been also found for lattice models with arbitrary Chern numbers that are built using the KM model [41]. By diagonalizing the Hamiltonian $H = PH_{\text{int}}P$ of the modified KM model at filling $\nu = 1/2$ on the torus with $H_{\text{int}} = \sum_i :n_i n_i:$, where $P$ is the projection operator to the lowest flatband and $::$ denotes normal ordering, we find that the two lowest energies are no longer exact zero modes as shown in Fig.4(a). When looking at the ground state degeneracy splitting for different system sizes, we find that such splitting is no longer zero for most system sizes as indicated in Fig.4(b). While this modified KM model with the constant Berry curvature still displays excellent ground state degeneracy, it's less ideal than the original KM model with the non-flat Berry curvature in this regard.

To further corroborate these ideas, we study the interacting two-body problem in the original and modified KM model. Any rotationally and translationally invariant interaction potential $V(\mathbf{r})$ can be decomposed in terms of the Haldane psuedopotentials [4, 42]. In the lowest Landau level, they read

$$V(\mathbf{r}) = \int d^2\mathbf{q} \sum_n v_n L_n(\mathbf{q}^2) e^{-q^2/2} e^{i\mathbf{q}\cdot\mathbf{r}}, \tag{8}$$

where $v_n$ are the psuedopotential parameters. The $\nu = 1/2$ bosonic Laughlin states are the densest zero-energy eigenstates of such Hamiltonian for contact interactions $v_n = \delta_{n,0} v_0$. The two-body spectrum for contact interactions in the continuum has only one non-zero constant energy at each center of mass momentum $\mathbf{K} = (\mathbf{k}_1 + \mathbf{k}_2)/2$, $E(\mathbf{K}) = v_0$.

Moving on to the lattice, the two-body spectrum is no longer guaranteed to be constant. There exist more than one non-zero energy in the two-body spectrum that depend on the center of mass momentum [43]. In the limit of a large unit cell, approaching the Landau level continuum, the two-body spectrum on the lattice approaches the continuum one (albeit with the difference that the number of finite levels per sector differs corresponding to the lower symmetry, hence fewer sectors, of the lattice system) [44]. The number of non-zero energies per momentum sector in the two-body spectrum is bounded from above by the number of non-zero eigenvalues of the interaction Hamiltonian. For on-site interactions, this number is hence bounded by the number of sites in the unit cell. The existence of two exact zero modes at half filling for the KM model with on-site interactions, on the other hand, implies that there is a maximum of two non-zero two-body energies per total momentum sector. This is indeed the case as shown in Fig 4(c). However, we find that the modified KM model exhibits an extra non-zero two-body energy per total momentum sector, implying a slight deviation from the ideal KM model that has only two non-zero two-body energies irrespective of the number of bands.

While we have demonstrated that flattening the curvature does not always imply more ideal FCI states, it is indeed beneficial to do this for certain models. We apply the optimization algorithm to the Hofstadter model with flux $\phi = 1/4$ per plaquette to obtain a modified Hofstadter model with constant Berry curvature. With on-site interactions and at half filling, we find that the modified Hofstadter model with the constant Berry curvature exhibits a two-fold ground state degeneracy with smaller energies and smaller ground state splitting than the original model as indicated in Fig 5(a-b). While both models have more than two non-zero two-body energies per total momentum sector as shown in Fig 5(c), we find that the *extra* non-zero two-body energies are smaller for the modified model (c.f the inset of Fig 5(d)). In this case, flattening the Berry curvature does make the Hofstadter model more *ideal* in the sense of having smaller energies (closer to zero) and smaller ground state splitting at half filling with on-site interactions in addition to having smaller *extra* two-body non-zero energies per total momentum sector. This is in agreement with the results of Ref. [31] that correlates the stability of FCI models with Berry curvature fluctuations. Our results suggest that the number of non-zero two-body energies per momentum sector could be a good a measure for the ideality of an FCI model while the Berry curvature fluctuations, by themselves, are generally not.

# 6 No ideal flatbands with constant curvature in lattice systems

As we saw in the previous section, making the curvature of the KM model constant does not always improve its properties in the FCI phase. Here we investigate the effect of other "ideal" band geometry conditions on the FCI physics, and their relation to the constant Berry curvature

condition. Following Refs. [19, 38] we call a QH liquid in a band with

$$4 \det g(\mathbf{k}) = \mathcal{F}(\mathbf{k})^2 \tag{9}$$

an *a ideal droplet*. This condition, together with $\det g(\mathbf{k}) \neq 0$, can equip the BZ with a Kähler structure pulled back from its image in the complex projective plane $\mathbb{C}P^{n-1}$ [39]. If the stronger condition

$$2g_{\mu\nu}(\mathbf{k}) = \delta_{\mu\nu} |\mathcal{F}(\mathbf{k})| \tag{10}$$

is satisfied, we talk about an *a ideal isotropic droplet*. Ref. [45] uses a slightly weaker constraint to define an *ideal flatband*:

$$2g_{\mu\nu}(\mathbf{k}) = \omega_{\mu\nu} |\mathcal{F}(\mathbf{k})|, \tag{11}$$

where $\omega$ is a constant, unit determinant positive definite matrix. This condition is equivalent to the previous one after an appropriate affine reparametrization of $\mathbf{k}$-space and gives rise to Bloch wave functions that are holomorphic functions of $k_x + ik_y$.

In order to quantify the deviation from the ideal flatband condition (11) with constant $\omega_{\mu\nu}$, we compute the standard deviation of $\omega(\mathbf{k})_{\mu\nu} = 2g(\mathbf{k})_{\mu\nu}/|\mathcal{F}(\mathbf{k})|$ over the BZ, summed over all components. This quantity is lowered by the flattening procedure in the Hofstadter model, but is increased in the KM model. Comparing the average third-highest 2-body energy, and the finite-size splitting of the ground state, we find that these properties of the interacting system are correlated with the degree of deviation from (11), and not the flatness of the Berry curvature, see Fig. 6.

In the rest of this section we show that it is not possible to simultaneously satisfy the ideal flatband condition (11) and have constant Berry curvature in any lattice system that has a finite number of degrees of freedom per unit cell.

We use the result of Ref. [18], which proves that condition (11) together with $\mathbf{k}$-independent $\mathcal{F}$ (hence $g$) implies that the density operators obey the generalized GMP, or $W_\infty$ algebra:

$$\left[\bar{\rho}_{\mathbf{q}}, \bar{\rho}_{\mathbf{q}'}\right] = 2i \sin\left(\frac{\mathcal{F}\epsilon_{\mu\nu}q_\mu q'_\nu}{2}\right) e^{g_{\mu\nu}q_\mu q'_\nu} \bar{\rho}_{\mathbf{q}+\mathbf{q}'}, \tag{12}$$

where $\bar{\rho}_{\mathbf{q}} = Pe^{i\mathbf{q}\mathbf{r}}P$ is the projected density operator with $P = \sum_{\mathbf{k}} |\mathbf{k}\rangle \langle\mathbf{k}|$ the projector onto the lowest Chern band and $\mathbf{r}$ the position operator.

In a lattice system with a single site per unit cell, $\bar{\rho}_{\mathbf{q}}$ is Brillouin zone periodic, $\bar{\rho}_{\mathbf{q}} = \bar{\rho}_{\mathbf{q}+\mathbf{G}}$ for reciprocal lattice vectors $\mathbf{G}$. If there are multiple sites per unit cell, but the orbital coordinates are rational linear combinations of the lattice vectors, the BZ can be extended such that $\bar{\rho}_{\mathbf{q}}$ is periodic with respect to the extended BZ. Substituting $\mathbf{q} \to \mathbf{q} + \tilde{\mathbf{G}}$ in (12) shows that this periodicity is incompatible with the density algebra, completing the proof by contradiction. In Appendix B we extend the proof to the case of irrational coordinates.

We note that the Kapit-Mueller model is a system with a finite number of degrees of freedom that has an ideal flatband satisfying (11). However, it does not pose a counterexample to our theorem, because the curvature is not constant for any finite flux per unit cell. The deformation of the Hamiltonian $H'(\mathbf{k}) = H(\mathbf{f}(\mathbf{k}))$ described in Sec. 3 preserves the weaker ideal droplet condition (9), however, in general it does not maintain condition (11) for general $\mathbf{f}(\mathbf{k})$, hence our modified KM model no longer has an ideal flatband, as we illustrate in Fig. 6. Condition (11) is equvalent to the quantum geometric tensor $\eta(\mathbf{k})$ having a constant null vector $|w_0\rangle$. We calculate the overlap of the approximate null vector $|w(\mathbf{k})\rangle$ of $\eta(\mathbf{k})$ with the exact null vector for the KM model $|w_0\rangle = (1, i)$. We see that the KM model has a constant null vector of $\eta$ to high precision, while the optimized model's null vector shows fluctuations.

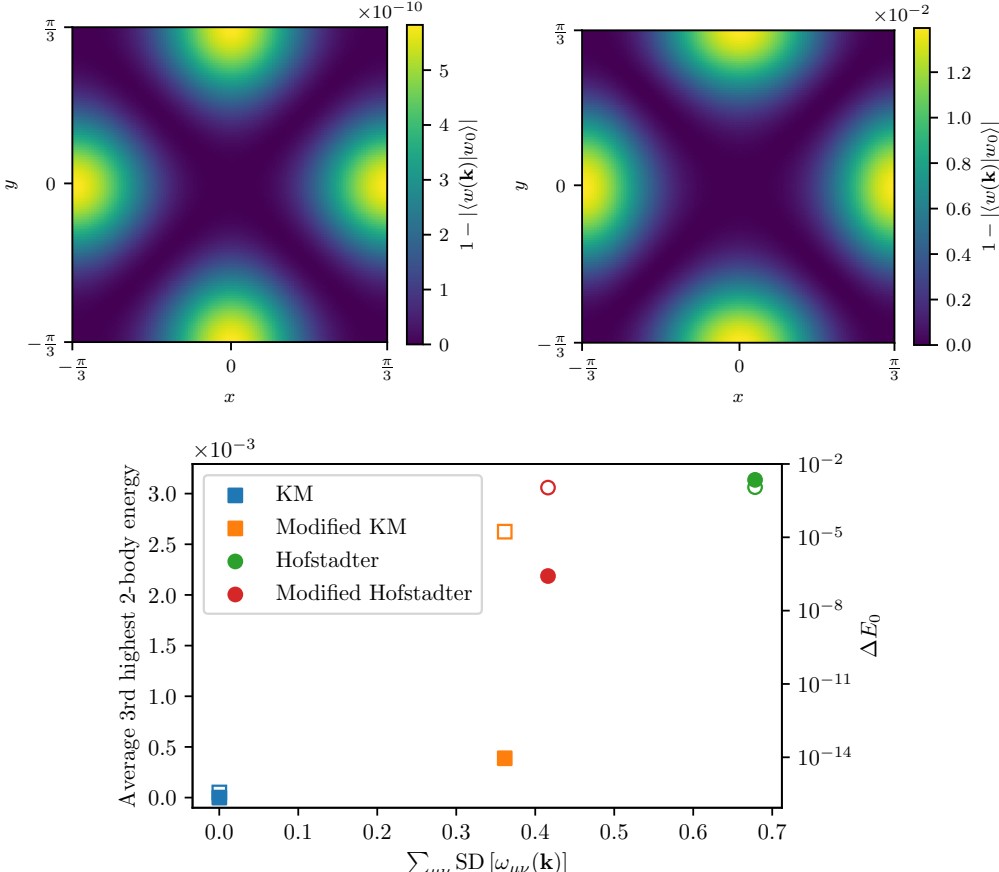

Figure 6: Above: Overlap of the null vector $|\omega(\mathbf{k})\rangle$ of the quantum geometric tensor with the exact null vector for the 3-band KM model (left), and optimized KM model with constant curvature (right). Note the scales of the colorbars. Below: Dependence of the average third highest 2-body energy (full symbols) and the average ground-state splitting (empty symbols) on the deviation from the ideal condition for the quantum metric.

# 7 Conclusion

In this manuscript we have studied the question whether constant Berry curvature, similar to Landau Levels, is possible in Chern bands. We answered in the affirmative, providing an explicit construction for systems with at least 3 orbitals per magnetic unit cell. Next we investigated the properties of bosonic fractional quantum Hall states in bands with constant curvature. We found that in the interacting case minimizing the curvature fluctuations does not necessarily result in properties that imitate LLs better. Instead we found that the ideal flat-band condition (11) (satisfied by the Kapit-Mueller model) determines the interacting physics, specifically the rank of the 2-body problem with on-site interactions, the number exact zero energy eigenstates per momentum sector. Finally we proved that it is not possible to have an ideal flatband with constant curvature satisfying the GMP algebra for density operators in a lattice model.

Our results indicate that it is necessary to go beyond the level of single-particle physics in order to better understand the connection between FCIs and FQHE. While constant curvature gives the identical algebra for the projected coordinates in FCIs and FQHE, it does not always improve the many-body spectra. The many-body properties in FQHE are captured by Hal-

danes's pseudopotentials [42]. In lattice models, both rotational and translational symmetries are broken. It is known that the model FQHE states and their pseudopotentials can naturally adapt to the breaking of rotational symmetry [34, 46–49]. In comparison, the discrete translational symmetry of FCIs leads to a different number of (two-body) states per momentum sector [43] and needs a more careful treatment.

Our results raise some open questions for future investigation. It is known that exactly flat bands are not possible with finite-range hoppings in lattice models [30]. Is it possible to prove an analogous statement about constant curvature, or ideal flatbands? While we showed that the GMP algebra (which follows from the ideal flatband condition with constant curvature) is not realizable in lattice models, we also conjecture that there is no nontrivial closed density algebra that lattice systems can admit; however, we do not have a rigorous proof of this statement. It is an interesting mathematical question, whether simultaneously constant curvature and metric are possible to satisfy globally (even without the ideal flatband condition). Put differently, we conjecture that a two-dimensional submanifold of $\mathbb{C}P^n$ with vanishing scalar curvature cannot be a torus with nonzero Chern number. Ref. [50] proposed a general formula for the Hall conductivity in interacting systems in terms of the Berry curvature and the momentum-dependent occupation number, however, this result remains controversial. [51,52] As the counterargument of Ref. [51] relies on non-constant curvature, our construction of flatbands with constant curvature can serve as a test case to elucidate this debate.

## Author contributions

E. J. B. posed the initial research question and goals, acquired funding, and oversaw the project. D. V. formulated the proof for the existence of band structures with constant curvature and the related numerical algorithm. D. V. and K. Y. proved the no-go theorems about constant curvature in two-band models and ideal flatbands. A. A. performed numerical calculations on FCI states. All authors took part in interpreting the results and writing the manuscript.

## Data availability

The source code generating all of the data shown in the figures is available at Ref. [53].

## Acknowledgments

D. V. is grateful to Gergő Pintér for helpful discussions. A. A. acknowledges helpful discussions with Zhao Liu. The authors are supported by the Swedish Research Council (VR) and the Wallenberg Academy Fellows program of the Knut and Alice Wallenberg Foundation.

## A   $C = 3$ model with constant curvature

We also apply the optimization algorithm to the three-band model with Chern number 3 of Ref. [38]. The resulting Berry curvature has relative variations of order $10^{-11}$, as shown in Fig. 7.

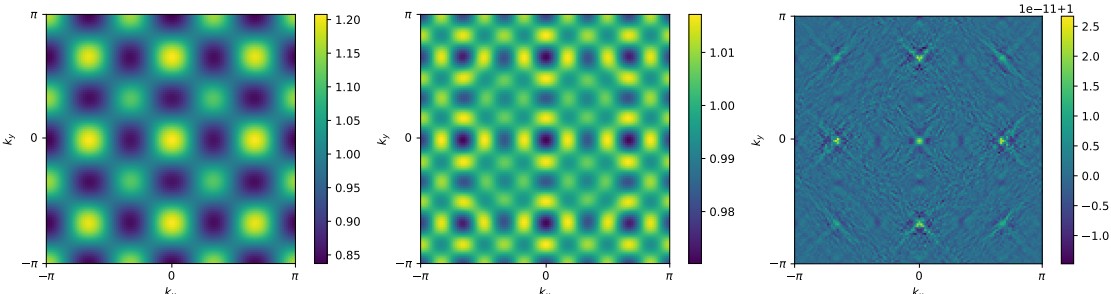

Figure 7: Berry curvature of the original 3-band model of Ref. [38] (left), after one iteration of the flattening algorithm (middle), and after 10 iterations (right). Note the scales of the colorbars. The curvature is scaled such that average curvature of 1 corresponds to a band with Chern number 3.

## B  Ideal flatband with constant curvature is not possible in a lattice model

Following Ref. [18], first we show that the projected density operator factorizes in an ideal flatband with constant curvature. The Fubini-Study metric and the Berry curvature have the relation

$$\text{tr } g(\mathbf{k}) = \langle \mathbf{k}|Pr_+Qr_-P|\mathbf{k}\rangle - \mathcal{F}(\mathbf{k}) = \langle \mathbf{k}|Pr_-Qr_+P|\mathbf{k}\rangle + \mathcal{F}(\mathbf{k}), \tag{13}$$

where $Q = 1 - P$ and $r_\pm = x \pm iy$. The operators $Pr_-Qr_+P = (Qr_+P)^\dagger Qr_+P$ and $Pr_+Qr_-P = (Qr_-P)^\dagger Qr_-P$ are positive semi-definite. For simplicity, we assume that $\mathbf{k}$-space is parametrized in a way, such that the isotropic ideal droplet condition $2g_{\mu\nu} = \delta_{\mu\nu}|\mathcal{F}|$ is also satisfied with constant $g$ and $\mathcal{F}$. For positive $\mathcal{F}$, this implies $Qr_+P = 0$. From this we can deduce $r_+P = Pr_+P$, and taking its adjoint, $Pr_- = Pr_-P$. Now writing the projected density operator

$$\begin{aligned}
\bar{\rho}_\mathbf{q} &= P\exp(i\mathbf{q}\cdot\mathbf{r})P \\
&= P\exp\left(\frac{i}{2}q_+r_-\right)\exp\left(\frac{i}{2}q_-r_+\right)P \\
&= \exp\left(\frac{i}{2}q_+Pr_-P\right)\exp\left(\frac{i}{2}q_-Pr_+P\right) \\
&= \exp(i\mathbf{q}\cdot Pr P)\exp\left(-\frac{\mathcal{F}\mathbf{q}^2}{4}\right),
\end{aligned} \tag{14}$$

where $q_\pm = q_x \pm iq_y$. We used the previous identities to propagate the band projector all the way into the power series from the left and right, and in the last step used the Baker-Campbell-Hausdorff formula and the commutation relation of the projected position operators $[PxP, PyP] = -i\mathcal{F}$. This factorization immediately implies that the GMP algebra is satisfied. The result for negative $\mathcal{F}$ is similar and we only need to replace $\mathcal{F}$ by $|\mathcal{F}|$.

On the other hand, writing the projected density operator in terms of the Bloch wavefunctions we find

$$\bar{\rho}_\mathbf{q} = \sum_\mathbf{k} u^\dagger_{\mathbf{k}+\mathbf{q}} u_\mathbf{k} |\mathbf{k}+\mathbf{q}\rangle\langle\mathbf{k}|. \tag{15}$$

As the $\mathbf{k}$-space translation operator

$$\sum_\mathbf{k} |\mathbf{k}+\mathbf{q}\rangle\langle\mathbf{k}| \propto \exp(iP\mathbf{q}\cdot\mathbf{r}P) \tag{16}$$

up to a complex phase, if (14) is satisfied then

$$F(\mathbf{q}) \equiv \left| u_{\mathbf{k}+\mathbf{q}}^{\dagger} u_{\mathbf{k}} \right| = \exp\left( -\frac{|\mathcal{F}||\mathbf{q}^2}{4} \right) \tag{17}$$

independent of $\mathbf{k}$. For rational site coordinates, the periodicity of the Bloch wavefunctions with respect to the extended BZ implies that $F(\tilde{\mathbf{G}}) = 1$ for all $\tilde{\mathbf{G}}$ extended reciprocal lattice vectors. This is incompatible with (17), providing an alternate proof for the case with rational site coordinates, which we extend to the irrational case in the following.

Let us assume that for some irrational site coordinates $F(\mathbf{q})$ satisfies (17). We can simultaneously approximate all the $x$ coordinates (and separately the $y$ coordinates) of the sites, and deform the positions to their rational positions without changing any of the onsite or hopping parameters in the tight-binding Hamiltonian. (Here for simplicity we assume a unit square unit cell, but the same argument is applicable with arbitrary unit cell shape writing the positions as linear combinations of the primitive lattice vectors.) Such a deformation of the coordinates by $\tilde{\mathbf{r}}_i = \mathbf{r}_i + \Delta\mathbf{r}_i$ changes the Bloch wavefunctions as $\tilde{u}_{\mathbf{k},i} = \exp(i\mathbf{k} \cdot \Delta\mathbf{r}_i) u_{\mathbf{k},i}$, but does not change the energy spectrum and leaves the Chern number invariant. Because $u_{\mathbf{k}}$ is normalized, the resulting change in $F(\mathbf{q})$ is bounded from above as

$$\Delta F(\mathbf{q}) \leq \max_i |\mathbf{q} \cdot \Delta\mathbf{r}_i| \,. \tag{18}$$

The n-dimensional version of the Dirichlet approximation theorem states that there are infinitely many denominators $p_x \in \mathbb{Z}$ such that the error in the rational approximation of all $x$ coordinates with fractions $m_i/p$ is bounded by

$$\left| \frac{m_i}{p_x} - r_{ix} \right| = |\Delta r_{ix}| \leq \frac{c}{p_x^{(1+1/n)}} \,, \tag{19}$$

where $n$ is the number of degrees of freedom in the unit cell and $c$ is some constant [54]. The same applies to the $y$ coordinates.

On the other hand, for extended reciprocal lattice vectors (17) gives

$$\Delta F(\tilde{\mathbf{G}}) = 1 - \exp\left( -\frac{|\mathcal{F}|\tilde{\mathbf{G}}^2}{4} \right) \,. \tag{20}$$

Choosing sufficiently large denominator $p_x$ and accurate approximation, substituting $\tilde{\mathbf{G}} = p_x \mathbf{G}_x$ (an extended reciprocal lattice vector in the $x$ direction) we get

$$\Delta F(p_x \mathbf{G}_x) \leq \frac{c}{p_x^{1/n}} \tag{21}$$

leading to a contradiction with (20) and completing the proof.

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
