# Peer review of "Topological Lattice Models with Constant Berry Curvature"

_SciPost Physics, doi:SciPost Phys. 12, 118 (2022)_

## Round 1 · Referee Report · Anonymous (Referee 1) · 2021-10-13

Strengths

This paper has studied the possibility of flatenning the Berry curvature and the energy bad in a topological band structure and its impact on the stability of fractional Chern insulators. This is definitely an interesting question.

Weaknesses

The paper is not well-written. The order of section is not chosen properly. The results seem contradictory (because of poor presentation).

Report

This paper has studied the possibility of flatenning the Berry curvature and the energy bad in a topological band structure and its impact on the stability of fractional Chern insulators (FCIs). They show that a uniform Berry curvature in the 1st Brillouin zone does not necessarily lead to a more stable FCI. They have also shown that an local lattice model with perfectly flat bands and exactly uniform Berry curvature cannot exist. However, if we compromise and allow a small variation in the Berry curvature or flatness of the energy band the model can indeed be realized on a lattice.

This main subject definitely of great ineterest for the researchers in the field of fractional quantum Hall effect (FQHE). However, the paper lacks a consistent flow and is not well written. The order of section is not chosen properly. The results seem contradictory (because of poor presentation).

For example, the authors discuss extensively the interplay between flat band structure and a flat Berry curvature and their impact on the stability of the FCIs. Later on, they argue such band structures cannot exist. This is quite confusing for the readers. Moreover, they discuss that such models are not as good as the regular Kapit-Muller model of 1/3 FQH state, and more stable than its 1/4 counterpart. A proper explanation is lacked in my opinion.

Finally, a lot of equations have been used without elucidating their origin and importance. Since their is no page limit in this journal, they need to expand those sections to make the article more accessible and self-contained.

Requested changes

The order of some section must change. The author should discuss their no go theorem first and more clearly emphasize that relaxing the exactly flat energy or exactly uniform Berry curvature is the loophole to their no go theorem.

They should also provide a better and more convincing explanation of their rather contradictory results on the stability of the FCIs in their nearly inform model.

The paper clearly needs a substantial revision and become more available and more appealing to meet the criteria and standards required by Scipost.

  • validity: good
  • significance: good
  • originality: high
  • clarity: low
  • formatting: reasonable
  • grammar: good

Author:  Daniel Varjas  on 2022-02-25  [id 2246]

(in reply to Report 1 on 2021-10-13)
Category:
answer to question
validation or rederivation

We thank the Referee for reviewing our manuscript, and finding that we studied an interesting research question. We regret that the Referee found the presentation of the results hard to follow. We made several changes, also highlighted in the attached manuscript.

The order of some section must change. The author should discuss their no go theorem first and more clearly emphasize that relaxing the exactly flat energy or exactly uniform Berry curvature is the loophole to their no go theorem.

We changed the order of the sections in order to clarify the message of the paper, and provided an overview of the paper’s structure in the last paragraph of the introduction. Now we first present the positive result about the existence of constant curvature lattice models, and only after this present the no-go theorem about constant curvature in 2-band models. We made sure to clarify that there are two no-go theorems in the manuscript, the second one shows that the ideal droplet condition (which we find to govern the FCI physics instead of the fluctuations of the Berry curvature) cannot be satisfied simultaneously with constant curvature in any lattice model.

They should also provide a better and more convincing explanation of their rather contradictory results on the stability of the FCIs in their nearly inform model.

We clarified our arguments as to why certain models benefit from flattening the Berry curvature, and others do not. We added an analysis in sec VI. where we quantify the deviation from the strictest ideal flatband condition (11), and we find that this is the relevant quantity for the interacting physics. This quantity is improved by the flattening in the Hofstadter model, but is made worse in the KM model. This is clear intuitively, as the KM model has wavefunctions that are already ideal by construction, and cannot be improved. We added a new panel to fig. 6, showing that this quantity is a good predictor of both the third highest 2-body energy’s deviation from zero, and of the size of the finite-size ground-state splitting.

Attachment:

Constant_curvature_diff_rvutAfQ.pdf

Anonymous on 2022-03-25  [id 2323]

(in reply to Daniel Varjas on 2022-02-25 [id 2246])
Category:
answer to question

The authors have properly addressed my questions and concerns and I support its publication.

---

## Round 1 · Referee Report · Anonymous (Referee 2) · 2022-2-8

Report

This paper address the question of finding lattice models possessing bands with exactly constant (non-vanishing) Berry curvature. The findings of the paper regarding this question can be divided into two major results. First, they show that roughly speaking it is possible to have exactly flat Berry curvature in models with more than two internal degrees of freedom per unit cell (three or more bands). As a complement to this finding, the authors prove a no-go theorem for the case of two-band models stating that in any two-band model, the berry curvature does vary within the Brillouin zone. Second, they show that albeit the possibility of constructing lattice models with constant Berry curvature, this property does not generally improve the topological robustness of the system, namely in a fractional Chern insulator model. This second finding is, in fact, against the initial motivation in pursuit of such constant-Berry-curvature models.

The aforementioned findings are given in detailed analytical proofs for the absence (existence) of the possibility of finding constant berry curvature in two (three or more) band models. Also, considering certain models, it is shown how to obtain the constant Berry curvature besides exploring the stability of FCI states under such conditions. In particular, the authors start with lattice models, introduced by Kapit and Mueller, whose ground states can be tailored to be exactly identical with Laughlin states. Then they apply their Berry curvature flattening recipe to obtain a modified Kapit-Mueller model with constant Berry curvature. Finally, they show by numerical inspections that although the modified Kapit-Mueller model still possesses the ground state degeneracy, it is less stable than the original Kapit-Mueller model with varying Berry curvature.

My overall assessment, based on the findings and presentation of the paper is positive, and after addressing my following concerns, I can suggest its publication.

  1. I find the introduction a bit technical which for a general audience of SciPost Phys. So I suggest extending/modifying the introduction to make it more accessible for non-expert readers.

  2. The second finding of the stability of FCI states under the constant Berry curvature condition is mainly supported by numerical calculations. It would be much more helpful if the authors can provide an intuitive physical picture for these findings as well. For instance, provide some explanation why in the case of the Hofstadter model with flux φ = 1/4 per plaquette, the flattening does act in favor of a more ideal model, but not in the case of φ = 1/3 Kapit-Mueller model.

  • validity: good
  • significance: good
  • originality: high
  • clarity: high
  • formatting: excellent
  • grammar: perfect

Author:  Daniel Varjas  on 2022-02-25  [id 2245]

(in reply to Report 2 on 2022-02-08)
Category:
answer to question
validation or rederivation

We thank the Referee for the positive review of our manuscript and the constructive criticism. We answer the two points below, and attach the updated manuscript with changes highlighted.

I find the introduction a bit technical which for a general audience of SciPost Phys. So I suggest extending/modifying the introduction to make it more accessible for non-expert readers.

Following the referee’s suggestion, we rephrased the third and fourth paragraphs of the introduction to make it more accessible, and to better highlight the structure of the manuscript.

The second finding of the stability of FCI states under the constant Berry curvature condition is mainly supported by numerical calculations. It would be much more helpful if the authors can provide an intuitive physical picture for these findings as well. For instance, provide some explanation why in the case of the Hofstadter model with flux φ = 1/4 per plaquette, the flattening does act in favor of a more ideal model, but not in the case of φ = 1/3 Kapit-Mueller model.

We thank the Referee for pointing out that the analysis of the numerical results was not sufficiently clear. We added an analysis in sec VI. where we quantify the deviation from the strictest ideal flatband condition (11), and we find that this is the relevant quantity for the interacting physics. This quantity is improved by the flattening in the Hofstadter model, but is made worse in the KM model. This is clear intuitively, as the KM model has wavefunctions that are already ideal by construction, and cannot be improved. We added a new panel to fig. 6, showing that this quantity is a good predictor of both the third highest 2-body energy’s deviation from zero, and of the size of the finite-size ground-state splitting.

Attachment:

Constant_curvature_diff.pdf

---

## Round 2 · Author Response

We thank the Referees for reviewing our manuscript, and the generally positive reactions. Both referees requested clarifications of the presentation of the manuscript, so we rephrased the introduction and changed the order of sections, see the detailed list of changes below. The referees also asked to clarify our interpretation of the numerical data, as a response we added further analysis to sec VI and Fig 6. We are confident that with these changes our manuscript is sufficiently clear and meets the standards of SciPost Physics.

---

## Round 2 · List of Changes

1. We rephrased the last two paragraphs of the Introduction, and added an overview of the structure of the paper, so it is easier to follow.
  2. We merged sections IV and V about the constant curvature models, and moved section III with the 2-band no-go theorem after them.
  3. We added further analysis of the relation of the ideal droplet condition and the FCI physics in section VI and figure 6.
  4. Changes of phrasing, corrections, and clarifications throughout the manuscript.

---

## Editorial Decision

published